# Uptake, Translocation, and Yield Assessment of Ca, K, S, and Fe in Three Potato (*Solanum tuberosum* L.) Cultivars (Agria, Désirée, and Red Lady) Grown Under Varying Soil Types

**DOI:** 10.3390/plants14091351

**Published:** 2025-04-30

**Authors:** Ana R. F. Coelho, Manuela Simões, José Almeida, Fernando H. Reboredo, Joaquim Cawina, Fernando Lidon

**Affiliations:** 1NOVA School of Sciences and Technology, Earth Sciences Department, Campus de Caparica, 2829-516 Caparica, Portugal; mmsr@fct.unl.pt (M.S.); ja@fct.unl.pt (J.A.); fhr@fct.unl.pt (F.H.R.); j.cawina@campus.fct.unl.pt (J.C.); fjl@fct.unl.pt (F.L.); 2GeoBioTec Research Center, NOVA University Lisbon, 2829-516 Caparica, Portugal

**Keywords:** *Solanum tuberosum* L., mineral translocation, uptake efficiency, yield assessment

## Abstract

Potato (*Solanum tuberosum* L.) is the world’s fourth most important food crop and is considered a staple food. Nutrient absorption in potato plants is influenced by different factors such as soil properties (namely mineral element composition). This study aimed to assess Ca, K, S, and Fe dynamics in three potato cultivars (Agria, Désirée, and Red Lady) grown across different soil types. As such, soil analyses were carried out before and after cultivation, revealing different variations in Ca, K, S, and Fe content. The results demonstrated that mineral distribution in the different plant organs (leaves, stems, roots, and tubers) showed distinct patterns, with Ca accumulating preferentially in leaves, K in stems, S in roots, and Fe in leaves. Mobilization efficiency was also evaluated and followed a specific pattern across all cultivars and soil types: K > S > Fe > Ca, reflecting the mineral translocation efficiency of these mineral elements within plant tissues to tuber. No significant differences were observed in tuber weight, or minimum and maximum diameter, indicating that these quality parameters were not influenced by the soil type. Fresh weight of tuber biomass assessment showed variability among the different cultivars and soil types. Principal component analysis showed that Ca content is associated with productivity parameters as well as K, contributing to tuber yield. Overall, cultivar-based variations in mineral uptake patterns were identified, suggesting that both genetic and environmental factors play an important role in mineral element absorption and translocation efficiency. This study highlights the importance of understanding mineral element dynamics in *S. tuberosum* L. cultivation, in order to optimize tuber yield and quality.

## 1. Introduction

Potato (*Solanum tuberosum* L.), originating from the Andean region of South America, is the world’s fourth most important food crop and ranks third in terms of human consumption, and thus regarded as a staple food [1,2,3].

Potatoes are valued for its nutritional profile rich in carbohydrates, proteins, vitamins, starch, and potassium (K) [4,5]. However, the concentrations of protein, starch, K, iron (Fe), zinc (Zn), and other mineral elements have variations among potato cultivars. Their content is influenced by different pre-harvest factors (e.g., environment or biotic and abiotic stresses) and post-harvest factors (e.g., storage or transport) [5].

Vegetatively propagated though seed tubers or whole potatoes [6,7], *S. tuberosum* L. development is influenced by environmental factors, such as temperature, water supply, period of light, and soil properties, which may affect and influence tuber formation, development, and even the number of tubers produced [7,8,9,10] and the impact nutrient availability and absorption [8,9,10].

Despite the significant commercial importance of different cultivars, such as Agria, Désirée, and Red Lady, comprehensive comparisons of mineral uptake and translocation patterns remain limited. These cultivars differ in growth, maturity timing, and yield potential, potentially leading to distinct patterns of mineral element dynamics. Understanding these cultivar-specific differences is crucial for optimizing cultivation practices.

Plants require 17 essential elements for normal growth and development [7,11,12], most of which are available in the soil [12], except for carbon (C), hydrogen (H), and oxygen (O), which are obtained from the atmosphere and from soil water [11]. Nitrogen (N), phosphorus (P), potassium (K), calcium (Ca), sulfur (S), iron (Fe), zinc (Zn), magnesium (Mg), manganese (Mn), copper (Cu), boron (B), molybdenum (Mo), chlorine (Cl), and nickel (Ni) [7] are either obtained from soil composition or supplemented through fertilizers [11].

Both nutritional quality and productivity of potato tubers are strongly influenced by mineral nutrition, particularly by elements such as Ca, K, or S, which play critical roles in plant growth, development, and stress responses [7]. In potato plants, each mineral element has distinct physiological functions in plant growth and development.

Potassium (K^+^), classified as a primary nutrient [7], is present in large quantities in potato plants [13,14], and impacts tuber number per plant, tuber size, yield, and storage quality [7,13]. It is crucial for enzyme activation, osmotic regulation, photosynthate translocation, photosynthesis, and carbohydrate formation, specifically in starch formation and translocation from leaves to tubers [13,14].

Calcium (Ca^2+^), a secondary nutrient [7], maintains cell wall structure and membrane stability, and functions as a second messenger in various physiological processes, particularly in plants’ response to biotic stresses [15,16,17]. Calcium deficiencies are associated with various tuber quality disorders, emphasizing its importance in crop development [7].

Sulfur (uptake as SO_4_^−^), another secondary nutrient [7] is essential for chlorophyll synthesis, photosynthesis efficiency, and in various metabolic processes and pathways [18]. It also acts as a signaling molecule (as sulfur and sulfur-containing compounds) in conditions of plant stress responses [19]. Previous studies have demonstrated that S influences tuber quality and yield [20,21]. In fact, S is considered one of the mineral elements that can limit plant yield and quality [22].

Iron (Fe) is considered an important micronutrient in plants, especially in chlorophyll synthesis and electron transport, as well as in different metabolic processes [23].

The efficiency with which these minerals are absorbed from the soil and translocated to various plant organs, particularly tubers [24], varies significantly among cultivars and can substantially impact both yield parameters and nutritional value. Mineral translocation ratios between source tissues (roots, stems, and leaves) and sink tissues (tubers) [25] provide valuable insights into the efficiency of nutrient partitioning within the plant, considering that each mineral element has different mobility within the plant.

Previous studies have demonstrated considerable variation in mineral uptake and utilization efficiency among potato cultivars [24,25,26,27]. However, the relationship between mineral translocation efficiency and productivity parameters such as tuber size distribution and yield has not been thoroughly characterized for these commercially important cultivars such as Agria, Désirée, and Red Lady under controlled conditions.

In this framework, our research aims to characterizes and compare the uptake and translocation patterns of Ca, K, S, and Fe in the vegetative organs of three cultivars of *S. tuberosum* L. (Agria, Désirée, and Red Lady) highly produced in Portugal and to correlate these patterns with the mineral composition of different types of soil. Moreover, this study also aims to evaluate the relationship between Ca, K, S, and Fe and yield parameters (tuber weight, caliber, and yield), and also seeks to identify cultivar-specific mineral translocation and their accumulation in plants’ tubers. These findings will contribute to the understanding of mineral use efficiency in potato and may lead to the implementation of strategies to optimize tuber yield and quality. As such, based on these aims, we address the following specific research questions: How do the uptake and translocation patterns of Ca, K, S, and Fe differ among Agria, Désirée, and Red Lady cultivars when grown in different soil compositions? What is the relationship between mineral translocation efficiency and yield parameters across the three cultivars? And does the initial mineral composition of soil influence the final mineral accumulation in different plant organs, particularly in tubers?

## 2. Results

### 2.1. Ca, K, S, and Fe Content in the Different Soils

The analysis of Ca, K, S, and Fe in the different soils (Table 1) revealed variations in their contents before plantation and after tuber harvest. In the “before plantation” data, the cultivars had not yet been introduced; however, the soil had already been divided into the different pots, and planting was subsequently carried out. Overall, Ca, K, S, and Fe concentrations were higher before plantation than after harvest. Nevertheless, in some cases (e.g., Ca content in Siro Horta, Type 1 and Composit 2 soils in cv. Agria), this pattern was not observed.

Considering the different soils, Ca content ranged from over 5000 mg/kg to less than 17,500 mg/kg, K varied between 5000 mg/kg and up to 15,000 mg/kg, S ranged from more than 1250 mg/kg to less than 8750 mg/kg, while Fe showed the highest variation, from 2500 mg/kg to as much as 40,000 mg/kg. These data highlight the variability in soil composition before and after harvest (see Figure 1).

Principal component analysis (PCA) was carried out (Figure 2) and illustrates the distribution of the soil samples before plantation (bp) and after harvest (ah) of the three potato cultivars. The first two principal components (Dim1 or PC1 and Dim2 or PC2) explain 52.7% and 17.3% of the total variance, respectively.

The separation observed along PC1 suggests that this axis is the main factor differentiating soil composition before plantation and after harvest, as well as among the three cultivars. This component is strongly associated with Fe_bp, Cu_bp, Pb_bp and Fe_ah.

On the other hand, PC2 is mainly influenced by K_bp, Ca_bp, and Hg_bp, suggesting that this component differentiates between the soil composition before plantation and after harvest, as well as between the three cultivars. Moreover, the higher contribution of Ca_bp and K_bp suggests that soils with higher initial content of these mineral elements may be associated with the future occurrence of different cultivar responses.

Among the three cultivars, distinct clustering patterns are observed, with Agria and Désirée showing a significant separation along PC2, indicating that the soils associated with these cultivars had different initial compositions and responded differently to the cultivation process (after harvest). The Red Lady cultivar exhibits a more compact distribution, suggesting lower variability in soil composition before plantation and after harvest.

Furthermore, the correlation between Pb_ah and Hg_ah suggests that both mineral elements may have been less affected by plant uptake, remaining in the soil after harvest across all the cultivars.

In summary, the PCA analysis indicates that the cultivation of the three cultivars influenced soil composition in distinct ways, possibly due to differences in nutrient absorption efficiency. Additionally, the strong correlation of Fe, Cu, and Pb with PC1 suggests that these mineral elements play a significant role in soil differentiation, whereas the initial Ca and K contents are a key factor in PC2.

Considering specifically Ca, K, S, and Fe and their respective contribution values (loadings), several observations can be derived from Figure 2.

For instance, Ca_bp exhibits a strong association with PC2, while Ca_ah shifts toward negative values, indicating a reduction in the Ca content of the soil after harvest, which may suggest significant plant uptake or leaching processes. K_bp displays a high loading in PC2, but, like Ca, it shifts toward negative values, suggesting that K levels in the soil also decreased following cultivation. On the other hand, S_bp is strongly correlated with PC1, yet S_ah occupies a different position compared to its initial state, indicating a reduction in its soil concentration after harvest. Meanwhile, Fe_bp demonstrates a strong loading in PC1 and retains a similar position in the plot after harvest, suggesting relative stability in its soil content.

### 2.2. Calcium Translocation in S. tuberosum L. Organs

There were significant differences between the different plant organs (tuber, root, stem, and leaf) of *S. tuberosum* L. considering the different soils (Figure 3). Regardless of the soil, a consistent trend of Ca accumulation was observed across the three cultivars, following the order: leaf > stem > root > tuber.

The lowest Ca content in roots was 5040 mg/kg in the Red Lady cultivar grown in Composit 2 soil, while the highest value was 14,037 mg/kg, also in Red Lady cultivar but in Type 3 soil. In tubers, the lowest Ca content was 1312 mg/kg in Agria cultivar, whereas the highest value was 2347 mg/kg in the Red Lady cultivar, both in Composit 2 soil.

The highest Ca concentrations were observed in stems (20,072 mg/kg) and leaves (37,538 mg/kg) in the Red Lady cultivar grown in Composit 1 soil. Meanwhile, the lowest Ca content in stems was 9160 mg/kg in the Red Lady cultivar, and in leaves it was 15,047 mg/kg in the Désirée cultivar, both in Type 1 soil.

### 2.3. Potassium Translocation in S. tuberosum L. Organs

Potassium contents were significantly higher in the stems across all three cultivars (Figure 4), while significantly lower concentrations were observed in the roots (except for soil Type 2 in Désirée and Red Lady cultivars). In general, K content in leaves and tubers was similar, although in most cases K content was lower in tubers compared to leaves.

The lowest and highest K content in roots was obtained in the Agria cultivar, with 21,407 mg/kg in Type 3 soil and 54,778 mg/kg in Composit 1, respectively. In tubers, the lowest K content was 39,614 mg/kg in the Red Lady cultivar grown in Composit 2, while the highest was 49,498 mg/kg in the Désirée cultivar on Type 3 soil. In stems, the highest K content was obtained in the Désirée cultivar in Composit 1 (112,269 mg/kg) and the lowest in the Agria cultivar in Type 3 soil (58,314 mg/kg). In leaves, the highest K content was 59,514 mg/kg in the Désirée cultivar in Composit 3 and the lowest was 21,020 mg/kg in the Désirée cultivar in Composit 1.

### 2.4. Sulfur Translocation in S. tuberosum L. Organs

Considering sulfur content (Figure 5) significant differences between the different organs were observed. Moreover, with the exception of soil Type 1 in all three cultivars, in all other analyzed soils, S content in the roots was higher than in the leaves. In our data, an outlier was observed in soil Composit 3 for the Agria cultivar, where S content follows the unique pattern of: root > stem > leaf > tuber.

Overall, S contents in stems and tubers in the three cultivars tended to be relatively similar.

The highest S content obtained in roots was 6877 mg/kg in the Red Lady cultivar in Composit 2, while the lowest was 2559 mg/kg in the Désirée cultivar in Type 1 soil. Considering S content in tubers, the highest content was 1925 mg/kg in the Red Lady cultivar in Type 1 and the lowest was 1009 mg/kg in the Agria cultivar in Composit 2 soil. In stems, the highest S content was 4787 mg/kg in Agria in Composit 3 (which is an outlier in our data) and the lowest was 862 mg/kg in the Red Lady cultivar in Siro Horta. In leaves, the highest S content was 4477 mg/kg in the Red Lady cultivar in Type 1 and the lowest was 1993 mg/kg in the Agria cultivar in Type 3 soil.

### 2.5. Iron Translocation in S. tuberosum L. Organs

Considering Figure 6, Fe content in some cases (e.g., in Composit 2 and 3 in the three cultivars), shows the highest concentration in the roots. However, in general, leaves tend to have the highest Fe content, followed by roots, stems and tubers. Across all the soils analyzed and the three cultivars, the lowest Fe contents were consistently observed in stems and tubers.

The highest Fe content was obtained in Composit 2 in roots (3966 mg/kg, in the Red Lady cultivar), in tubers (393 mg/kg, in the Désirée cultivar), and in stems (493 mg/kg, in the Red Lady cultivar). The lowest Fe contents in roots, tubers, and stems were 163 mg/kg in the Red Lady cultivar in Siro Horta, 51 mg/kg in the Agria cultivar in Siro Horta, and 94 mg/kg in the Désirée cultivar in Type 1, respectively. Regarding Fe content in leaves, the highest was 877 mg/kg in the Agria cultivar in Composit 1 and the lowest was 386 mg/kg in the Désirée cultivar in Type 1 soil.

### 2.6. Mobilization Efficiency of Ca, K, S, and Fe in Tubers

The mobilization efficiency of Ca, K, S, and Fe in tubers is represented in Figure 7, where the ratios for all the mineral elements analyzed are greater than 1. However, the ratios of the mineral elements analyzed are different, indicating different mobilization efficiencies to the tubers, also being an indicator of the different mobility within the plant. In this context, it is also possible to verify that each soil within the three cultivars showed different mobilization efficiencies.

Regarding Ca, it showed the lowest ratios across the three cultivars, while the highest ratio among the different soils was observed in Type 1 soil. Overall, K exhibited higher ratios compared to Ca, S, and Fe. Moreover, a general trend of similarity in ratios among the three cultivars was observed for S and Fe, particularly between the Désirée and the Red Lady cultivars.

Considering Ca and K, a greater variability in the mobilization efficiency was observed in Composit 1 and 2 compared to the other soils.

In this context, there appears to be a general trend in mobilization efficiency following the order of: K > S > Fe > Ca.

### 2.7. Yield Parameters

Tuber weight, minimum and maximum diameter, and fresh weight of tuber biomass were analyzed, considering the different soils and cultivars (Table 1). No significant differences were observed between soils within each cultivar, except for the minimum diameter in the Red Lady cultivar, where a significantly higher value was recorded in Type 3 soil and a significantly lower value in Type 1.

Average tuber weight ranged from 43.87 g to 63.22 g, 34.67 g to 59.90 g, and 34.42 g to 51.57 g, respectively, for the Agria, Désirée, and Red Lady cultivars.

Regarding minimum diameter, the values ranged from 4.03–4.80 cm, 3.95–4.57 cm, and 2.67–4.03 cm, respectively, for the Agria, Désirée, and Red Lady cultivars. Considering the Agria, Désirée, and Red Lady cultivars, respectively, the parameter of maximum diameter, the values ranged from 4.74 cm to 6.30 cm, 4.17 cm to 6.00 cm, and 3.77 cm to 5.70 cm. Thus, Agria cultivar showed the highest average tuber weight and minimum and maximum diameter, followed by Désirée and Red Lady cultivars.

Regarding fresh weight of tuber biomass, Agria, Désirée, and Red Lady showed, respectively, a variation among the different soils of 171.8–323.7 g/plant, 164.6–257.8 g/plant, and 59.7–308.2 g/plant. In this context, Agria and Désirée cultivars showed the highest fresh weight of tuber biomass on Type 3 soil, while Red Lady cultivar exhibited the highest fresh weight of tuber biomass in Composit 3.

### 2.8. Principal Component Analysis

Considering the PCA analysis performed with all the analyzed parameters (Figure 8) it is possible to observe different distributions. For instance, the cultivars are distributed in distinct areas of the plot, suggesting that each cultivar differs in the way it absorbs and translocates mineral elements.

Tuber weight, minimum and maximum diameter, and yield (fresh weight of tuber biomass) are grouped in the same direction, indicating a positive correlation among these parameters, all of which are related to plant productivity.

Regarding Ca in roots and leaves (Ca_root and Ca_leaf) they are positioned in the same region as the parameters related to productivity, suggesting an association. As such, this indicates that Ca may play an important role in crop productivity. However, tubers do not show significant Ca accumulation compared to the other plant organs.

On the other hand, Fe, Pb, and As are positioned relatively far from the productivity parameters, indicating that their presence does not appear to be associated with plant productivity.

The Agria cultivar exhibits a stronger association with productivity parameters and calcium accumulation. In contrast, the Désirée cultivar appears more dispersed, indicating greater variability in mineral absorption.

Calcium and K exhibited distinct translocation patterns, whereas in general Ca tends to accumulate more in the leaves rather than in tubers, and K in general is more associated with stems.

Moreover, S and Fe exhibit more variable mobilization patterns, and the Désirée and Red Lady cultivars appear to have a more homogeneous absorption pattern of S compared to Agria cultivar. Additionally, the Red Lady and Désirée cultivars seem to be more strongly associated with heavy metal accumulation.

Overall, significant differences in mineral absorption and translocation are observed among cultivars, indicating distinct patterns in the use of mineral elements by plants.

## 3. Discussion

The soil analysis revealed significant variations in Ca, K, S, and Fe content before and after plantation; however, overall, their content before plantation was higher than after plantation (except in Ca in Siro Horta, type 1 and Composit 2 soils in cv. Agria) (Figure 1). The variation observed is probably due to some problem in sample homogeneity or even due to the different dynamics of absorption and translocation of these mineral elements by plants [29]. For instance, the variations in soil content observed align with the fact that potato cultivation can change soil nutrient profiles due to the high nutrient demand from this crop, especially P and K [22]. Regarding Fe content, the data show a wide range (Figure 1), suggesting a more pronounced impact on Fe dynamics in soil, probably due to oxidation and reduction processes [30] which can occur depending on cultivation conditions. The higher uptake of K by plants verified in our data (Figure 1) is in accordance with Koch et al. (2020) [22] which mentioned that potatoes can uptake substantial amounts of K from soil. Moreover, the results of Ca, K, S, and Fe content in soil are higher compared to the typical agricultural soils, due to the soil from Canal Caveira, an area known for its mining history and higher mineral content.

The literature indicates that the initial content of Ca and K in soil plays an important role in soil differentiation [31], which aligns with the PCA results (Figure 2), where Ca and K are associated with the PC2. Moreover, considering Pb and Hg content in soil, the PCA results indicate that both mineral elements remained in soil after harvest being less translocated to plants, which is in accordance with Sanderson et al. (2019) [32] findings which indicate that the total concentration of Cd (a heavy metal) in soil is not always a good indicator of its concentration in the edible parts of plants, even in contaminated soils. On the other hand, the decrease of Ca, K, S, and Fe could indicate a strong plant uptake or even leaching or precipitation processes, as discussed in previous research on the mobilization of mineral elements in soils by plants [33], which could explain the observed changes in soil after plantation.

Considering Ca, K, S, and Fe content in the different organs of *S. tuberosum* L., different distribution patterns were observed (Figure 3, Figure 4, Figure 5 and Figure 6). Calcium, for instance, showed a higher accumulation in the leaves, followed by stems, roots, and tubers (Figure 3) regardless of soil type, which was already previously documented in the literature in which Ca is often transported to the leaves and other organs, while its accumulation in tubers tends to be more limited with a higher rate of xylem needed [29]. This distribution pattern aligns with the pattern of Ca low phloem mobility and primary transport through the xylem [29,34,35,36] and the higher Ca content in leaves compared to tubers, reflects Ca physiological role in cell wall stabilization and membrane integrity in photosynthetic active tissues [35,37].

Potassium showed a greater accumulation in the stems across the three cultivars analyzed (Figure 4), being consistent with its role in metabolic regulation in plants [29], especially as the K^+^ functions as the primary osmotic active cation in the translocation of assimilates through the phloem [15]. Indeed, it was previously reported that vegetative parts of potato plants contain more K than tubers [38], being in accordance with our data. In fact, K plays a crucial role in phloem translocation of assimilates and can be used to overcome local energy limitations [15], being highly mobile in phloem and its higher content in stems (Figure 4) reflects its role in growth of new tissues, enzyme activation, and protein synthesis.

Sulfur showed a unique pattern (Figure 5) compared to Ca and K, where the highest content of S was obtained in roots, followed by stems, leaves, and tubers. This accumulation pattern of higher content in roots followed by stems, is probably due to their important role in amino acid and protein synthesis in both organs, being crucial for plant growth [22]. However, this pattern contradicts in part the findings in Agria, Picasso, and Rossi cultivars of *S. tuberosum* L. [39], where S content was higher in leaves, followed by roots. As such, the higher accumulation of S in roots (Figure 5) can be due to a lower translocation efficiency or even an increase in root retention mechanisms, probably as an adaptation to the different soil conditions in our study. Also, the highest accumulation in roots can also be due to the role of rhizosphere microbiota and mycorrhizal associations in S uptake.

Moreover, the unique pattern of S translocation in Agria cultivar (Figure 5) can represent and outlier, and can be probably to a cultivar-specific response in Composit 3 Soil, such as potential soil–cultivar interactions or unique physiological adaptations of Agria.

Iron content was overall, higher in leaves, followed by roots, stems, and tubers (Figure 6). The higher accumulation of Fe in leaves can possibly be due to its role in photosynthetic processes, especially in chlorophyll synthesis and electron transport [23]. Some exceptions in the distribution pattern of Fe in *S. tuberosum* L. organs can be due to the fact that soil composition significantly influences Fe mobilization and translocation in plants [40].

The ratio of Ca, K, S, and Fe (mobilization efficiency) was greater than 1, indicating that the mineral elements analyzed are essential for tuber growth, being stored in this *S. tuberosum* L. organ (Figure 7). Additionally, the mobilization efficiency of Ca, K, S, and Fe suggests that K has a dominant role across the three cultivars of *S. tuberosum* L. and soil types, followed by S and Fe. This accumulation pattern (K > S > Fe > Ca) is in accordance with the physiological mobility of these mineral elements within plants and considering soil composition, as previously mentioned by Etienne et al. (2018) [41]. For instance, K is one of the most efficiently translocated mineral elements, being very mobile in the phloem [42] and has a central role in establishing tubers and starch [22] and in translocating carbohydrates from leaves to tubers [43]. On the other hand, Ca showed a more restricted mobilization efficiency to tubers due to the xylem transport and limited redistribution via the phloem [29,34,35,36], as previously mentioned. As such, the mobilization efficiency obtained (Figure 7) aligns with the translocation pattern observed in Figure 3, Figure 4, Figure 5 and Figure 6.

The PCA plot (Figure 8) revealed that Ca content is associated with productivity parameters, suggesting that Ca is important in overall crop productivity despite having a limited accumulation in tubers, being supported by Palta (2010) [44] research, who referred to the fact that Ca enhances plant vigor, contributing to tuber yield. Additionally, K was also associated with crop productivity. As previously reported by our first research paper, Red Lady and Désirée are associated with higher heavy metal accumulation [28] (Figure 8), which suggests cultivar-based variations in mineral uptake patterns [45]. According to Figure 8, Agria associate more with Ca and yield (fresh weight of tuber biomass) than Désirée and Red Lady cultivars, which suggests underlying genetic and metabolic differences among these cultivars. In general, regarding the different soil formulations, no significant differences were observed in tuber weight or minimum and maximum diameter (Table 1), indicating that they were not influenced by the soil. However, it was possible to verify soil influence, with a different response across the different cultivars in fresh weight of tuber biomass. Moreover, the tuber yield (fresh weight of tuber biomass) obtained in our experiment cannot be directly extrapolated to predict precise field yields due to factors such as root restriction, but provides valuable insights how cultivars behave among the different soil conditions.

Considering our findings, it is possible to verify that there is a complexity in the interaction between mineral elements (especially Ca, K, S, and Fe) in the different soil types and the way they are absorbed, translocated, and accumulated in the different organs of *S. tuberosum* L., which varied across cultivars and which varied at the different growth stages of plants life cycle This suggests that genetic and environmental factors play a crucial role in mineral element absorption and translocation efficiency, as previously mentioned. Moreover, our study also verified an association of Ca and K with crop productivity (Figure 8), highlighting the importance of these mineral elements in plant growth and development, and the need for proper agricultural practices to optimize potato yield.

## 4. Materials and Methods

### 4.1. Experimental Design

The experimental trial was conducted at the Department of Earth Sciences at NOVA FCT (Almada, Portugal) in an outdoor space using 27 plastic pots (10 L of capacity, 23 cm of height and 12 cm of average radius) according to what is indicated in the study by Coelho et al. (2025) [28]. Different types of soils were utilized, namely Siro Horta brand, recommended for vegetable plants and mixes carried out with soil samples collected from Canal Caveira (soil composition data are presented in the study of Coelho et al. (2025) [28], as well as the sites of sample collection).

Seed potatoes of three *S. tuberosum* L. cultivars (Agria, Désirée, and Red Lady) were pre-germinated from 5 April to 5 May of 2023, in a humid and dark environment for four weeks before being planted in the pots. The experimental design is described in Coelho et al. (2025) [28] and no fertilizers were applied to the different soils

Planting took place on 5 May, and harvesting was carried out on 17 July and 18, 75 days after planting. The production cycle was shorter than expected due to the closure of the facilities in August. Throughout the experiment, no additional fertilizers were applied beyond what was already present in the soil.

A constant monitoring of the plant development status was carried out throughout the experimental period (Figure 9). Plants were checked for pest issues weekly through visual inspections; however, no pests were detected during the experiment.

### 4.2. Climate Conditions

Climate conditions of temperature, air humidity, and precipitation were obtained from Meteostat online platform (https://meteostat.net/, accessed on 20 March 2025) for Caparica, Portugal. During the experimental period (5 May to 18 July), the temperature ranged between 14.4 °C and 35 °C, and precipitation from 0 mm to 11.5 mm, with an average of 0.45 mm.

### 4.3. Mineral Content in Soil and S. tuberosum L. Organs

The mineral content of Ca, K, S and Fe were determined in the different soil formulations before plantation and after harvest, and also in *S. tuberosum* L. organs (tubers, root, stems, and leaves) after harvest. The mineral content of Cu, Pb, As, and Hg has already been described in the results of Coelho et al. (2025) [28]. For *S. tuberosum* L. organs, the analysis was performed using an XRF analyzer (VANTA^TM^ Handheld XRF Analyzer, Olympus, Espoo, Finland), following proper samples preparation. The samples were carefully washed with deionized water to remove any particles, then dried at 60 °C in a laboratory oven until reaching a constant weight. They were then ground into fine powder using a mechanical mill and analyzed in the equipment previously calibrated. The samples were analyzed at least in quadruplicate.

### 4.4. Mobilization Efficiency of Ca, K, S and Fe in Tubers

The mobilization efficiency (ME) of the mineral elements analyzed was calculated to assess the ability of the plants to translocate Ca, K, S, and Fe from the vegetative organs (roots, stems, and leaves) to the tubers. The ME was determined using the following Equation (1):(1)$$ME=\frac{\mathrm{Mineral}\;\mathrm{content}\;\mathrm{in}\;\mathrm{tubers}}{\mathrm{Mineral}\;\mathrm{content}\;\mathrm{in}\;\mathrm{roots}+stems+leaves}$$
where a ME ratio greater than 1 indicates accumulation of a certain mineral element in the tuber, while a ratio lower than 1 suggests higher accumulation in the roots, stems, or leaves.

### 4.5. Yield Parameters

At harvest, the different *S. tuberosum* L. tubers of each plant were carefully collected, and any adhering soil was removed. The fresh weight of individual tubers was measured using a digital balance with an accuracy of 0.01 g. The average weight of tubers was determined considering 3 to 4 tubers of each pot. The area of each pot was 0.26 m^2^.

Maximum and minimum diameters of each tuber were determined using a digital caliper with a precision of 0.01 mm. The maximum diameter was defined as the longest distance across the tuber, while the minimum diameter was assessed perpendicular to the maximum diameter. These measurements were carried out in triplicate or quadruplicate.

Fresh weight of tuber biomass was calculated per plant by summing the fresh weights of all tubers harvested from each pot. The fresh weight of tuber biomass was expressed in grams per plant.

### 4.6. Statistical Analysis

Statistical analysis was performed using R software (version 4.4.2). One-way ANOVA was applied to mineral content of the different soils to assess differences between before plantation and after harvest considering each soil for each cultivar (Agria, Désirée, and Red Lady). Additionally, one-way ANOVA was also used for mineral content to evaluate differences between each *S. tuberosum* L. organ (root, stem, leaf, and tuber) within each soil and was carried out for each cultivar. Tukey’s test was used for mean comparison and a 95% confidence level was adopted for all tests.

PCA analysis was performed for soil mineral content assessment before plantation and after harvest; the data from the two main principal components were plotted, as well as the data for all the different parameters analyzed (mineral content in soil, root, stem, leaf, tuber, and yield parameters).

## 5. Conclusions

This research provides comprehensive insights into the dynamics of essential mineral elements (Ca, K, S, and Fe) in *S. tuberosum* L. cultivation. Our data demonstrated distinct patterns of mineral translocation and distribution across the different plant organs (leaves, stems, roots, and tubers), namely a preferential accumulation of Ca in leaves, K in stems, S in roots, and Fe in leaves. The mobilization efficiency of Ca, K, S, and Fe showed a specific pattern across the different cultivars and soil types: K > S > Fe > Ca, which aligns with the phloem mobility of these mineral elements within the plant. Among the different soil types and cultivars, no significant differences were obtained in tuber weight, minimum and maximum diameter. The variability in yield among cultivars and soil types indicates that it is influenced by the interaction between genotypes and growing environment. Calcium and K content are the two mineral elements associated with productivity parameters in principal component analysis. Nevertheless, cultivar-specific variations in mineral uptake, translocation and distribution patterns were verified, emphasizing the importance of future research regarding the understanding of mineral element dynamics in *S. tuberosum* L. cultivation in order to optimize tuber yield and quality in a specific cultivar.

## Figures and Tables

**Figure 1 plants-14-01351-f001:**
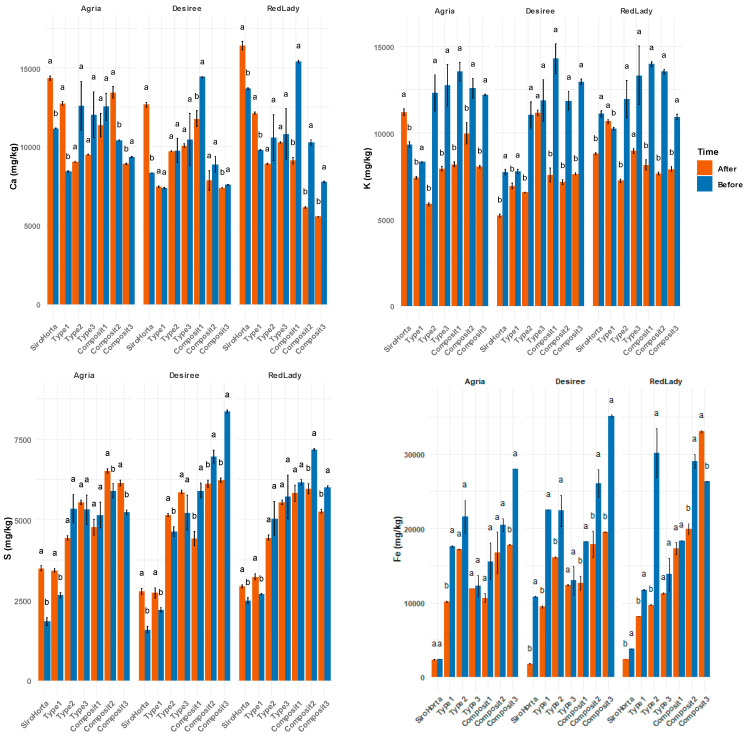
Calcium, K, S, and Fe average content (mg/kg) of soil (Siro Horta, Type 1, Type 2, Type 3, Composit 1, Composit 2, and Composit 3) before plantation (before) and after tuber harvest (after). Each value represents mean (*n* = 4–6) ± SE. ANOVA analysis, *p* < 0.05, performed for each soil and separately for each cultivar. Different letters express significant differences between each time of analysis (before harvest and after harvest) for soil (a, b), with a for the highest values.

**Figure 2 plants-14-01351-f002:**
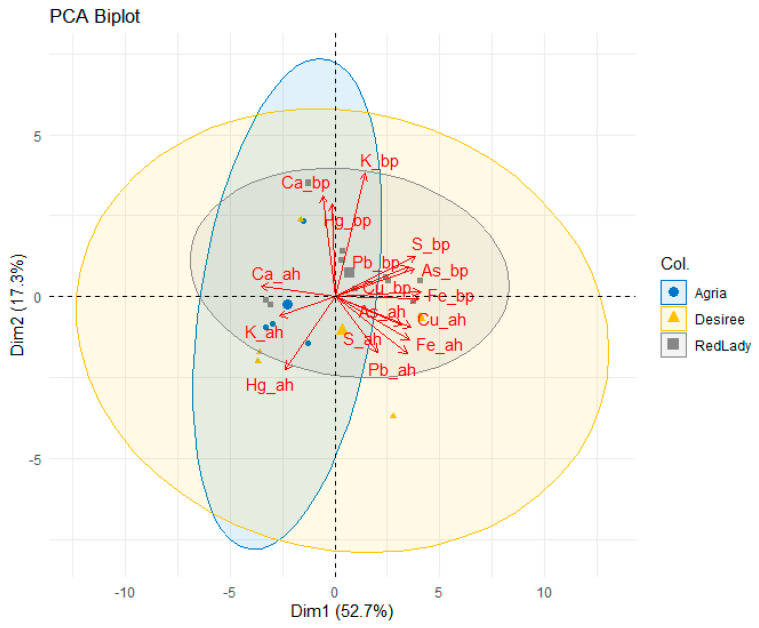
Principal component analysis (PCA) for PC1 and PC2 for each parameter studied (variance of 52.7% for PC1 and 17.3% for PC2) and representation and values of loadings of variables. As, Hg, Pb, and Cu data are presented in Coelho et al. 2025 [28].

**Figure 3 plants-14-01351-f003:**
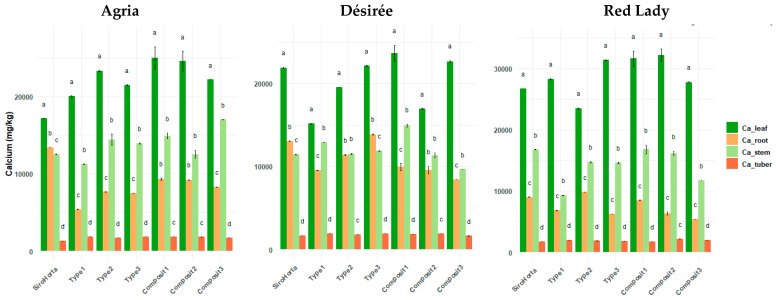
Calcium content (mg/kg) in *S. tuberosum* L. cvs. Agria, Désirée, and Red Lady organs (leaf, root, stem and, tuber) per soil (Siro Horta, Type 1, Type 2, Type 3, Composit 1, Composit 2, and Composit 3). Each value represents mean (*n* = 4–6) ± SE. ANOVA analysis, *p* < 0.05, performed separately for each cultivar, for each soil between each organ. Different letters express significant differences for each organ among each soil (a,b,c,d), with a for the highest values.

**Figure 4 plants-14-01351-f004:**
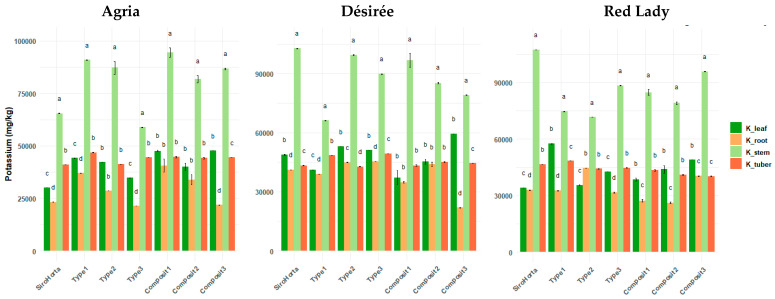
Potassium content (mg/kg) in *S. tuberosum* L. cvs. Agria, Désirée, and Red Lady organs (leaf, root, stem, and tuber) per soil (Siro Horta, Type 1, Type 2, Type 3, Composit 1, Composit 2, and Composit 3). Each value represents mean (*n* = 4–6) ± SE. ANOVA analysis, *p* < 0.05, performed separately for each cultivar, for each soil between each organ. Different letters express significant differences for each organ among each soil (a, b, c, d), with a for the highest values.

**Figure 5 plants-14-01351-f005:**
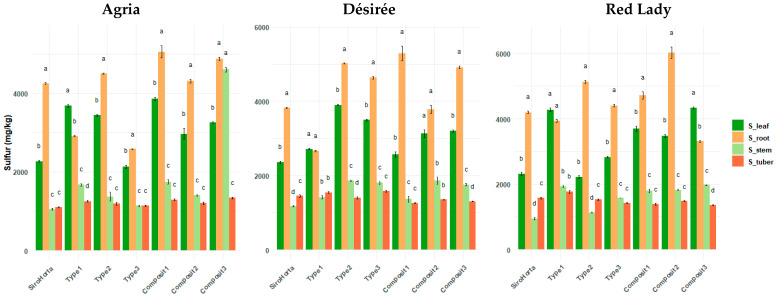
Sulfur content (mg/kg) in *S. tuberosum* L. cvs. Agria, Désirée, and Red Lady organs (leaf, root, stem, and tuber) per soil (Siro Horta, Type 1, Type 2, Type 3, Composit 1, Composit 2, and Composit 3). Each value represents mean (*n* = 4–6) ± SE. ANOVA analysis, *p* < 0.05, performed separately for each cultivar, for each soil between each organ. Different letters express significant differences for each organ among each soil (a, b, c, d), with a for the highest values.

**Figure 6 plants-14-01351-f006:**
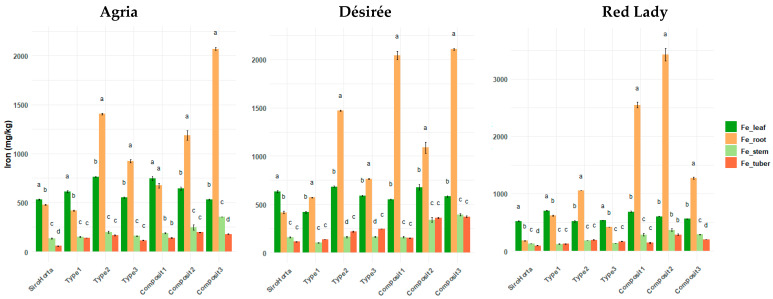
Iron content (mg/kg) in *S. tuberosum* L. cvs. Agria, Désirée, and Red Lady organs (leaf, root, stem, and tuber) per soil (Siro Horta, Type 1, Type 2, Type 3, Composit 1, Composit 2, and Composit 3). Each value represents mean (*n* = 4–6) ± SE. ANOVA analysis, *p* < 0.05, performed separately for each cultivar, for each soil between each organ. Different letters express significant differences for each organ among each soil (a, b, c, d), with a for the highest values.

**Figure 7 plants-14-01351-f007:**
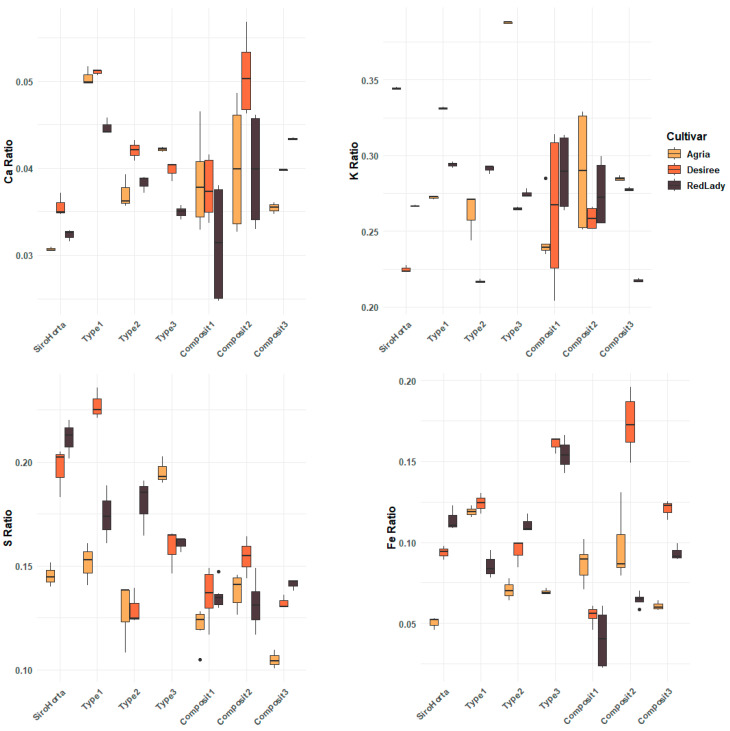
Mobilization efficiency index of Ca, K, S, and Fe in *S. tuberosum* L. cvs. Agria, Désirée, and Red Lady per soil (Siro Horta, Type 1, Type 2, Type 3, Composit 1, Composit 2, and Composit 3).

**Figure 8 plants-14-01351-f008:**
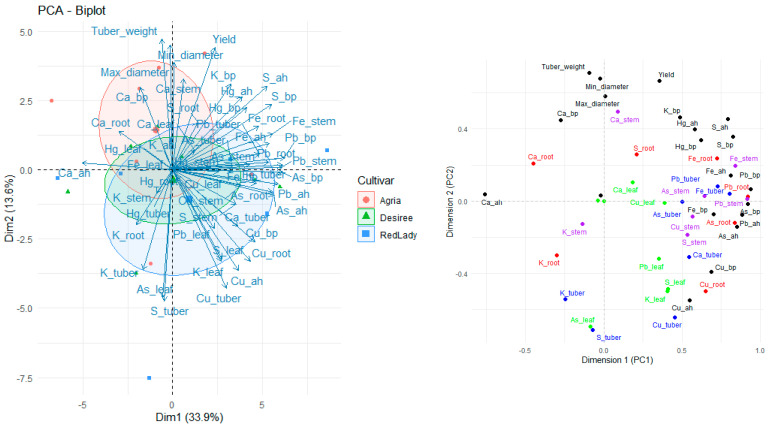
Principal component analysis (PCA) for PC1 and PC2 for each parameter studied (variance of 34.9% for PC1 and 13.1% for PC2) and representation of loadings of PC1 and PC2 of variables. As, Hg, Pb, and Cu data are presented in Coelho et al. 2025 [28].

**Figure 9 plants-14-01351-f009:**
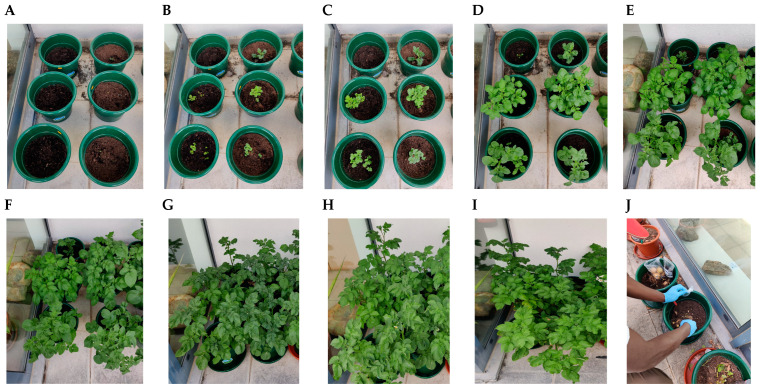
Example of monitoring of three cultivars of *S. tuberosum* L. throughout experimental period. Images illustrate different growth stages of crops. (**A**) 11 May; (**B**) 15 May; (**C**) 17 May; (**D**) 22 May; (**E**) 26 May; (**F**) 30 May; (**G**) 5 June; (**H**) 14 June; (**I**) 17 June; (**J**) 18 July.

**Table 1 plants-14-01351-t001:** Evaluation of tubers, regarding average weight (g), minimum diameter (cm), maximum diameter (cm), and fresh weight of tuber biomass (g/plant) in *S. tuberosum* L. cvs. Agria, Désirée, and Red Lady by soil type. Each value of average weight and minimum and maximum diameter represents mean and ± SE (*n* = 3–4), ANOVA analysis *p* < 0.05. Different letters express significant differences for each cultivar and soil among parameters (a,b), with letter a for the highest value.

Soil	Cultivar	Tuber Weight	Minimum Diameter	Maximum Diameter	Fresh Weight of Tuber Biomass
Siro Horta	Agria	55.40 ± 9.34 a	4.47 ± 0.29 a	5.00 ± 0.00 a	231.5
Type 1		51.47 ± 19.17 a	4.07 ± 0.37 a	5.57 ± 1.03 a	207.2
Type 2		62.00 ± 6.32 a	4.40 ± 0.12 a	6.30 ± 0.12 a	271.2
Type 3		59.23 ± 13.28 a	4.73 ± 0.35 a	5.73 ± 0.69 a	323.7
Composit 1		62.75 ± 26.11 a	4.15 ± 0.60 a	5.68 ± 1.18 a	249.7
Composit 2		63.22 ± 10.82 a	4.80 ± 0.34 a	5.60 ± 0.51 a	284.8
Composit 3		43.87 ± 14.78 a	4.03 ± 0.45 a	4.73 ± 0.58 a	171.8
Siro Horta	Désirée	47.40 ± 8.79 a	4.50 ± 0.36 a	4.53 ± 0.32 a	164.6
Type 1		35.97 ± 1.62 a	4.10 ± 0.21 a	4.17 ± 0.07 a	205.7
Type 2		59.90 ± 26.22 a	4.37 ± 0.28 a	6.00 ± 0.44 a	220.1
Type 3		59.57 ± 30.18 a	4.57 ± 0.88 a	5.23 ± 1.13 a	257.8
Composit 1		46.13 ± 4.37 a	4.25 ± 0.13 a	5.08 ± 0.26 a	230.2
Composit 2		46.53 ± 7.49 a	4.43 ± 0.35 a	4.98 ± 0.36 a	206.4
Composit 3		34.67 ± 5.08 a	3.93 ± 0.35 a	4.47 ± 0.23 a	250.5
Siro Horta	Red Lady	36.87 ± 7.55 a	3.23 ± 0.41 ab	5.70 ± 0.46 a	134.9
Type 1		36.17 ± 3.41 a	2.67 ± 0.03 b	3.83 ± 0.37 a	59.7
Type 2		28.00 ± 5.03 a	3.37 ± 0.15 ab	3.77 ± 0.29 a	167.5
Type 3		51.57 ± 20.28 a	4.03 ± 0.67 a	5.47 ± 0.74 a	183.2
Composit 1		34.42 ± 2.49 a	3.68 ± 0.15 ab	4.82 ± 0.27 a	247.05
Composit 2		44.80 ± 12.89 a	3.82 ± 0.20 ab	5.45 ± 0.63 a	200.9
Composit 3		46.70 ± 8.15 a	3.80 ± 0.15 ab	5.53 ± 0.49 a	308.2

## Data Availability

The original contributions presented in this study are included in the article. Further inquiries can be directed to the corresponding author.

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
