# Peer review of "Uptake, Translocation, and Yield Assessment of Ca, K, S, and Fe in Three Potato (Solanum tuberosum L.) Cultivars (Agria, Désirée, and Red Lady) Grown Under Varying Soil Types"

_plants, 2025, doi:10.3390/plants14091351_

Round 1
Reviewer 1 Report
Comments and Suggestions for Authors
Dear authors!
Thank you for submitting your article for review. It is devoted to the study of the absorption of Ca, K, S and Fe from the soil by three potato varieties during the growing season and the distribution of these elements in plant tissues. I really liked this work, its idea and execution. The data are subject to modern statistical processing. The literature referred to by the authors is modern, which emphasizes the relevance of the study.
There are some minor comments on the methodological part of the work:
- The caption to Figure 9 does not indicate the description of the figure under the letter "I".
- In Section 4.1. nothing is written about the agrotechnical methods of caring for plants during the growing season. In addition, it is unclear from the description under what conditions the experiment was conducted - in the field or in a greenhouse in pots?
After a little revision, the article can be successfully published in the journal "Plants".
Respectfully Yours, reviewer.
04 April 2025
Author Response
Suggestion of the reviewer: “I really liked this work, its idea and execution. The data are subject to modern statistical processing. The literature referred to by the authors is modern, which emphasizes the relevance of the study.”
Reply of the authors: The authors thanks you for your thoughtful review of our work. We are grateful for your positive feedback regarding both the concept and execution of our study.
Suggestion of the reviewer: “The caption to Figure 9 does not indicate the description of the figure under the letter “I””
Reply of the authors: We sincerely thank the reviewer for their careful examination of our manuscript and for pointing out this oversight. This was an error on our part, and we have now corrected the caption to include the missing description.
Suggestion of the reviewer: “In Section 4.1. nothing is written about the agrotechnical methods of caring for plants during the growing season. In addition, it is unclear from the description under what conditions the experiment was conducted - in the field or in a greenhouse in pots? ”
Reply of the authors: The authors thank the reviewer for their valuable observation. The authors have added more detailed information about the agrotechnical methods and experimental conditions as requested. We have now clarified that the experiment was conducted in an outdoor space using pots and have included specific details about fertilization approach, pest management procedures.
Reviewer 2 Report
Comments and Suggestions for Authors
attached

There are many grammatical errors that should be modified.
Author Response
Suggestion of the reviewer: “There are many grammatical errors in the whole article.. Some sentences are awkwardly structured (e.g., "Plays a crucial role in phloem translocation..." is missing a subject).. Carefully proofread for grammar, and ensure all sentences are complete and active… check and correct, please ”
Reply of the authors: The authors thank the reviewer for highlighting these grammatical issues. We have carefully proofread the entire manuscript to correct these issues, paying particular attention to ensuring all sentences are complete and properly structured. The revised manuscript has been thoroughly reviewed for grammar and clarity to improve the overall quality of the writing. We appreciate the reviewer's attention to detail, which has helped enhance the readability of our article.
Suggestion of the reviewer: “I suggest another title for example: (Mineral Dynamics and Yield Response in Three Potato (Solanum tuberosum L.) Cultivars Grown in Diverse Soil Compositions).
Or (Uptake, Translocation, and Yield Assessment of Ca, K, S, and Fe in Potato Cultivars under Varying Soil Types).
Reply of the authors: The authors appreciate the reviewer's suggestion regarding the title of our manuscript, and we agree that the proposed alternatives better capture the essence of our research. The title was changed to: Uptake, Translocation, and Yield Assessment of Ca, K, S and Fe in three potato (Solanum tuberosum L.) cultivars (Agria, Désirée and Red Lady) Grown under Varying Soil Types.
Suggestion of the reviewer: “The introduction doesn't explicitly highlight the knowledge gap early in the text. Only sees the gap in lines 94–101. so please move the explanation of the gap (cultivar-specific justify the study sooner. ”
Reply of the authors: As recommended, we have moved the knowledge gap explanation to an earlier position in the introduction.
Suggestion of the reviewer: “The first few paragraphs are quite general and can be shortened. Background on potato origin and propagation, while informative, distracts from the focus on mineral uptake and soil influence. could you summarize early background into 2–3 concise sentences, and shift focus more quickly to mineral nutrition and cultivar differences? ”
Reply of the authors: The authors, as recommended, have significantly condensed the general background information in the first few paragraphs of the introduction.
Suggestion of the reviewer: “The flow jumps between general crop information, physiology, mineral functions, and propagation without smooth transitions. try to group similar content and ensure logical progression (e.g., potato → nutritional importance → mineral requirements → cultivar differences → knowledge gap).” and “Some functions of K, Ca, and S are repeated differently. Please condense repetitive points and integrate them into concise, focused explanations of each mineral's role. ”
Reply of the authors: The authors have reorganized the introduction to follow a more logical progression and grouped similar content together to improve flow. We've also condensed the repetitive information about mineral functions into more concise explanations.
Suggestion of the reviewer: “Add 2–3 clear research questions or hypotheses at the end of the introduction to guide the reader. ”
Reply of the authors: The authors thank the reviewer for this valuable suggestion. Following the recommendation, we have added three clear research questions at the end of the introduction to better guide the reader through our study's objectives.
Suggestion of the reviewer: “In the results, the reported ranges for soil Ca (5000–17,500 mg/kg), K (5000–15,000 mg/kg), and Fe (2500–40,000 mg/kg) are extremely high compared to typical agricultural soils. No explanation is given for why these values are so elevated (e.g., contamination, fertilization, or analytical method bias). ”
Reply of the authors: We appreciate the reviewer's observation regarding the elevated mineral content values in our soil samples. The reviewer is correct that the reported ranges for Ca (5000–17,500 mg/kg), K (5000–15,000 mg/kg), and Fe (2500–40,000 mg/kg) are higher than typical agricultural soils.
These elevated values are primarily due to the soil compositions used in our study, which included commercial soil (Siro Horta) and mixtures with samples from Canal Caveira, an area known for its mining history and elevated mineral content. The commercial soil was specifically chosen for its enriched nutrient profile suitable for vegetable cultivation. We have added this explanation to the discussion section to provide context for these values.
Suggestion of the reviewer: “The "ratio" used for mobilization efficiency is undefined—is it (tuber content)/(root content)? A clear formula is needed. ”
Reply of the authors: The authors have presented the ratio for mobilization efficiency and the formula in section 4.4. of Materials and Methods.
Suggestion of the reviewer: “Ca, K, and S generally decrease after harvest, but some exceptions occur (e.g., Ca increases in some soils). No explanation is given (e.g., mineralization, experimental error). ”
Reply of the authors: The authors explained that in lines 353 to 354 of discussion section. In this section, we explain that these variations could be attributed to potential sample homogeneity issues or differences in mineral element absorption and translocation dynamics among the plants, as supported by previous research.
Suggestion of the reviewer: “In Fig. 5 (Sulfur translocation), Agria in Composit 3 shows an unusual pattern (root > stem > leaf > tuber), but no discussion explores why. ”
Reply of the authors: The authors thank the reviewer for highlighting this important observation. We have now expanded in our discussion the unusual sulfur translocation pattern (root > stem > leaf > tuber) observed in the Agria cultivar grown in Composit 3 soil. This addition provides potential explanations for this cultivar-specific response.
Suggestion of the reviewer: “Why does Ca accumulate more in leaves while K is highest in stems? No discussion of plant physiology (e.g., Ca immobility vs. K mobility in phloem). ”
Reply of the authors: The authors explained the Ca accumulation in S. tuberosum L. organs in the discussion part (lines 376- 384), while the K accumulation in S. tuberosum L. organs was completed in lines 389-393.
Suggestion of the reviewer: “Why does Agria associate with Ca and yield, while Désirée and Red Lady show more variability? No discussion of genetic or metabolic differences. ”
Reply of the authors: The authors expanded on these findings in the discussion section.
Suggestion of the reviewer: “Despite PCA suggesting Ca and K influence productivity (Fig. 8), no direct statistical correlation (e.g., regression) is shown between soil nutrients and tuber yield. ”
Reply of the authors: While PCA indicated an association between Ca, K, and productivity (Fig. 8), we acknowledge the reviewer's suggestion to present direct statistical correlations (e.g., regression) between soil nutrients and tuber yield. However, given the complex interactions among nutrients, soil types, and cultivars in our experimental design, PCA was initially chosen as an exploratory tool to identify major trends and relationships within the dataset. Future studies could explore multiple regression models to address multicollinearity, non-linear regression to identify optimal nutrient ranges, and cultivar-specific regression analyses. Despite the lack of a direct statistical correlation in our analysis, the PCA results (Fig. 8) and the existing literature (Palta, 2010) strongly suggest that Ca and K play important roles in overall crop productivity.
Suggestion of the reviewer: “In the Discussion. While the discussion highlights elemental dynamics (Ca, K, S, Fe), it lacks a thorough correlation with soil pH, texture, organic matter content, cation exchange capacity (CEC), or redox potential, which significantly affect nutrient mobility and availability. Correlate elemental uptake with pH, CEC, organic carbon, and microbial activity to better understand soil-plant interactions. ”
Reply of the authors: The authors understand the reviewer point regarding the importance of correlating our elemental dynamics data with soil properties like pH, CEC, organic matter, and redox potential. These factors significantly influence mineral element mobility and availability, and that a more comprehensive analysis would strengthen our understanding of soil-plant interactions. In fact, we have already performed those detailed soil analyses (including pH, etc.), and those data are published in a separate article (Coelho et al., 2025). In this particular study, however, our primary focus was on understanding the dynamics of these mineral elements – specifically Ca, K, S, and Fe – in relation to uptake and distribution across the different organs of the potato plants, and how this varied among the three cultivars. While we recognize the value of directly linking these dynamics to the soil properties in this paper, we felt it was important to first address and understand the distribution of these minerals among the plant organs, and how it differs considering the soil type and across the three cultivars under study.
Suggestion of the reviewer: “Pot experiments provide controlled insights but may not fully mimic field conditions. Replicating the study in open-field trials would validate findings under real-world agronomic conditions ”
Reply of the authors: The authors completely agree with the reviewer that pot experiments, while providing valuable controlled insights, may not fully represent real-world field conditions. The reviewer is right that replicating this study in open-field trials would be an excellent way to validate our findings under more realistic agronomic conditions. That is definitely a direction we hope to pursue in the future. Our intention with the current study was to first carefully examine and characterize the differences in mineral uptake and partitioning among these cultivars under controlled conditions. We see this work as a necessary first step before moving to more complex field trials. We acknowledge that field validation is a crucial next step for translating our findings.
Suggestion of the reviewer: “There is no mention of temperature, moisture regime, or irrigation practices, all of which could influence nutrient mobility and uptake, especially in pot experiments that can deviate from field realism.”
Reply of the authors: Details regarding temperature, moisture regime, and irrigation practices are indeed included in the Materials and Methods section. Specifically, the implementation of the trial was designed to align with typical potato planting practices in Portugal. We maintained daily irrigation, with the exception of weekends due to facility closures. The irrigation method/regime is also described in that section. We attempted to mimic Portuguese field conditions as closely as possible, with the understanding that this was a pot experiment. We are also aware of the limitations this brings, which is why we did not implement any phytosanitary treatment during the experiment.
Suggestion of the reviewer: “The role of rhizosphere microbiota or mycorrhizal associations in nutrient mobilization, particularly for Fe and S, is not addressed. These could be important modulators in elemental uptake. ”
Reply of the authors: The authors appreciate reviewer comment regarding the potential role of rhizosphere microbiota and mycorrhizal associations in nutrient mobilization, particularly for Fe and S. We add a sentence about it in the discussion.
Suggestion of the reviewer: “Although mineral element mobilization to tubers is discussed, there's no mention of post-harvest quality, nutritional implications, or shelf-life, which are important from an applied/agronomic perspective.”
Reply of the authors: The authors appreciate the reviewer's comment regarding the significance of post-harvest quality, nutritional implications, and shelf-life from an applied/agronomic perspective. As noted in the introduction, the established role of Ca in post-harvest quality has been previously examined. Furthermore, the emphasis of this research is not on the post-harvest qualities of the tubers, but rather on understanding the elemental dynamics of Ca, K, S, and Fe in the different organs of Solanum tuberosum grown in soil mixtures from a former mine (Canal Caveira). This focus is driven by the fact that these tubers are not intended for human consumption; instead, they serve as a means to investigate elemental uptake, distribution, and the impact on yield (related to the mineral elements uptake and influence) within the plant itself. As such, we have focused on what happens withing the plant. While future studies could indeed explore post-harvest aspects of potatoes grown in these soil conditions.
Suggestion of the reviewer: “Interactions or antagonistic effects between elements (e.g., Ca vs. Mg, Fe vs. Mn) are not considered, despite their potential to affect translocation efficiency. ”
Reply of the authors: The authors appreciate the reviewer pointing out that interactions or antagonistic effects between elements (e.g., Ca vs. Mg, Fe vs. Mn) that were not explicitly considered in our analysis, despite their potential to affect translocation efficiency. You are absolutely correct that these interactions are important and can influence nutrient uptake and distribution. Our focus in this study was primarily on the overall patterns of individual element translocation within the plant, rather than on the complex interplay between elements. We are currently developing a separate manuscript focused specifically on modeling these elemental interactions (synergies and antagonisms) and their impact, with the goal of evaluating the complex relationship with the mobilization efficiency.
Reviewer 3 Report
Comments and Suggestions for Authors
1.What is the basis for selecting soil types and potato varieties in this study? Generally speaking, the yield or appearance quality of potatoes of the same variety will be affected by soil type, and different genotype qualities may also have differences for the same soil type. However, this article does not fully explain the reasons for the insignificant differences.
2.The mineral translocation efficiency of potatoes varies at different growth stages and is influenced by other mineral elements, which is a complex process that cannot be explained clearly through an experiment. The author should supplement this in the discussion section.
The section on materials and methods is too brief, it is recommended to supplement it. For example, the basic physical and chemical properties of soil are presented in the form of a list for readers to understand.
3.Is it feasible to convert the tuber yield of potted plants into actual field yield? The author's consideration of only differences in area is not comprehensive.
4.Did the author apply base fertilizer to ensure the normal production of crops? Has fertilization been applied during the growth process? Does the application of these additional mineral elements affect the experimental factors?
Author Response
Suggestion of the reviewer: “What is the basis for selecting soil types and potato varieties in this study? Generally speaking, the yield or appearance quality of potatoes of the same variety will be affected by soil type, and different genotype qualities may also have differences for the same soil type. However, this article does not fully explain the reasons for the insignificant differences.”
Reply of the authors: The authors thank the reviewer for raising this important question regarding the basis for selecting soil types and potato varieties in our study. We understand your concern that the differences observed may stem from the choice of soils and cultivars. We did not aim to find significant differences among soils and cultivars, instead, we aimed at understanding the role of different soils in mineral element translocation to the different organs of Solanum tuberosum L. organs. We had a range of soil conditions commonly found in potato-growing regions of Portugal, especially close to mining regions. The chosen cultivars are commonly planted in Portugal and there is a lack of literature conserving those three cultivars. As the reviewer mentioned, it is indeed true that soil type and genotype interactions influence yield and quality, and we believe that the lack of differences observed across soil types reflects a balance of these effects. Additionally, with our research was possible to verify that genetic and environmental factors play a crucial role in mineral element absorption and translocation efficiency. Moreover, we could identify trends in elemental distribution patterns across cultivars, providing valuable insights into the underlying physiological processes regarding nutrient uptake and translocation. Future studies could certainly benefit from a more worldwide targeted selection of potato cultivars and soil types to maximize the potential for observing differences and exploring specific interactions in another depth.
Suggestion of the reviewer: “The mineral translocation efficiency of potatoes varies at different growth stages and is influenced by other mineral elements, which is a complex process that cannot be explained clearly through an experiment. The author should supplement this in the discussion section.
The section on materials and methods is too brief, it is recommended to supplement it. For example, the basic physical and chemical properties of soil are presented in the form of a list for readers to understand.”
Reply of the authors: The authors appreciate the reviewers’ suggestion regarding the complexity of mineral translocation in potatoes, which varies at different growth stages and is influenced by interactions with other mineral elements. We agree that this is a complex process that cannot be fully explained through a single experiment, and that is warrants further discussion. Our research aimed to evaluate that at harvest, but future research needs to be conducted thought the different growth stages of Solanum tuberosum L. plants. In this context, our study provides insights of mineral element distribution at a specific point in the potato plants development (harvest). However, we acknowledge that mineral translocation efficiency changes throughout the plants life cycle, as nutrient demands shift especially during vegetative growth, tuber initiation and tuber bulking. We incorporate a more thorough discussion of these complexities in the revised manuscript.
The authors did not provide the soil physical and chemical properties because it is already presented in a previous manuscript (Coelho et al., 2025), as mentioned in section 4.1. of Materials and Methods. Additionally, the authors supplemented the section materials and methods as suggested by the reviewer.
Suggestion of the reviewer: “Is it feasible to convert the tuber yield of potted plants into actual field yield? The author's consideration of only differences in area is not comprehensive.”
Reply of the authors: The authors thank the reviewer for raising the point about the feasibility of converting pot yield to actual yield, due to root restriction which can affect plant development. As such, our pot experiment provides and indication of the relative performance of different cultivars and soil types under controlled conditions, but it cannot be directly translated into precise field yield predictions. However, we believe that trends that were observed could provide insights which can be seen in mineral elements analysis and the way the genotypes behave in such conditions. The authors supplemented the section discussion regarding this matter.
Suggestion of the reviewer: “Did the author apply base fertilizer to ensure the normal production of crops? Has fertilization been applied during the growth process? Does the application of these additional mineral elements affect the experimental factors?”
Reply of the authors: The authors thank the reviewer for raising these questions regarding fertilization. We confirm that no fertilizers or phytosanitary products were applied before or during this experiment, a point which has been incorporated into the section material and methods of the manuscript. The experimental variables in this study were deliberately limited to soil type and cultivar. This approach was chosen to minimize factors and provide a clearer understanding of the inherent differences in mineral element uptake and mobilization under these specific conditions.
Round 2
Reviewer 2 Report
Comments and Suggestions for Authors
no more comments
Author Response
Dear reviewer, we thank you for all your suggestions and we appreciate the time and effort you've taken to provide feedback.
Best regards,
Reviewer 3 Report
Comments and Suggestions for Authors
The author has made careful revisions to the manuscript, and Fig 10 Temperature and Preparation can be considered for deletion.
In addition, Tab1 should be changed to a three line table,
the "Pot yield" should be suggested to be changed to "fresh weight of tuber biomass (g/plant)".
Author Response
Suggestion of the reviewer: “The author has made careful revisions to the manuscript, and Fig 10 Temperature and Preparation can be considered for deletion.”
Reply of the authors: The authors thank the reviewer for the feedback on our manuscript. We have deleted figure 10 as recommended.
Suggestion of the reviewer: “In addition, Tab1 should be changed to a three line table,”
Reply of the authors: The authors have reformatted to a three-line table format Table 1, as suggested by the reviewer.
Suggestion of the reviewer: “the "Pot yield" should be suggested to be changed to "fresh weight of tuber biomass (g/plant)".”
Reply of the authors: The authors have changed the terminology from “Pot yield” to “fresh weight of tuber biomass (g/plant)” as recommended by the reviewer.